# How Much Do We Care about Teacher Burnout during the Pandemic: A Bibliometric Review

**DOI:** 10.3390/ijerph19127134

**Published:** 2022-06-10

**Authors:** Valentina Gómez-Domínguez, Diego Navarro-Mateu, Vicente Javier Prado-Gascó, Teresa Gómez-Domínguez

**Affiliations:** 1Faculty of Education Sciences, International University of Valencia, 46002 Valencia, Spain; valentina.gomez@campusviu.es; 2Department of Specific Educational Needs and Attention to Diversity, Faculty of Education Sciences, Catholic University of Valencia, 46110 Valencia, Spain; diego.navarro@ucv.es; 3Department of Social Psychology, Faculty of Psychology, University of Valencia, 46010 Valencia, Spain; vicente.prado@uv.es

**Keywords:** burnout, stress, teachers, bibliometrics, COVID-19

## Abstract

In this study, a descriptive bibliometric analysis of the scientific production was performed in the Web of Science on burnout and/or stress in teachers in pandemic situations. The aim of the study was to analyse the scientific production on stress and burnout in teachers during the COVID-19 pandemic. A total of 75 documents from 33 journals with 3947 cited references were considered, with 307 researchers from 35 countries publishing at least one article. The country with the most publications was the USA, followed by China and Spain. The USA was the country with the most collaborations. A total of 184 institutions published documents, and the universities with the most records were Christopher Newport and Columbia, although the American University of Sharjah and Cape Breton University had a higher overall citation coefficient. Of the 33 journals that have published on the subject, *Frontiers in Psychology* and the *International Journal of Environmental Research and Public Health* stood out in terms of the number of articles, and they were also listed in this order with regard to their impact factor.

## 1. Introduction

The global pandemic caused by COVID-19 has caused a serious health emergency that has had significant global effects [1]. The rapid spread of the virus has also led to border closures, travel bans, the closure of businesses and workplaces and lockdowns of non-essential workers in many countries [2,3]

This situation has posed significant challenges for the population and has impacted on workers’ health, productivity and efficiency [4]. Along with the impact that a pandemic can have in itself, a key element is the perception of the pandemic, which can result in negative emotions such as uncertainty, fear, insecurity and stress. These emotions have an impact on performance at work [5,6].

One of the first measures that governments around the world adopted to help maintain social distancing and reduce infections was to close schools [4,7,8,9]. By 20 March 2020, schools in 137 countries had been closed as a result of COVID-19 [10,11]. School and university teaching went online and remained there throughout the 2010–2021 academic year [4,12,13].

Teachers have had to face new situations at work, as during lockdowns they had to adapt their teaching to the new pandemic context. Their work demands changed, as they had to adapt to online environments and the use of new teaching methodologies, which together with factors such as emotional exhaustion and a lack of resources led them to experience stress and burnout [9,14,15]. This exponential increase in teachers’ workload and efforts occurred without an equal increase in the control they could exercise or in the resources or rewards allocated to them, which, as postulated by Karasek’s demands–control model [16] and the effort–reward model [17], has led to a substantial increase in work-related stress [18,19,20]. Such an increase in work-related stress levels over time could be related to an increase in burnout symptoms [21,22,23,24,25].

Burnout syndrome refers to this chronification of work-related stress, manifesting as a long-lasting response to chronic emotional and interpersonal stressors at work [24,26]. As suggested by several authors [27,28], burnout seems to have serious consequences for both workers and the organizations to which they belong [29]. Among the former, a wide range of consequences have been described for the individual’s physical and mental health, including cardiovascular, respiratory, muscular, digestive and nervous system (anxiety, depression, etc.) disorders, and various psychosomatic alterations [30]. It sometimes manifests through maladaptive behaviours such as eating disorders, obesity, self-medication, alcohol and psychoactive substance use or marital and family conflicts [31,32,33,34,35,36,37,38]. From the point of view of organisations, the problems it creates include [39,40,41] deterioration in job performance and quality of work, low job satisfaction, absenteeism, increased job turnover, reduced interest and effort in performing work activities, increased interpersonal conflicts, increased workplace accidents and, of course, a decline in the quality of the individual’s working life [42,43,44,45], which seems to greatly affect business productivity [44,46,47].

In the specific case of teachers, burnout can negatively affect teaching effectiveness [48,49,50], teachers’ interactions with students [51], their work motivation [52], their ability to support students [53,54] and absenteeism [55]. It can also affect teachers’ health and well-being at work, increasing their likelihood of suffering from various pathologies such as depression [56] or insomnia [57], among others.

This situation has generated the need for research on how teachers have coped with this situation and the psychological states arising from it that have affected their teaching practice [44,45,46]. The psychological consequences of this change in the work environment include an increase in the number of hours worked from home, isolation and the lack of communication between colleagues, among others. These factors affect workers in general [58] and teachers in particular [59].

Given the importance of burnout and its likely increase due to the situation caused by the emergence of COVID-19, it is to be expected that there has been a high impact on this topic in the available scientific production in the scientific literature.

In this respect, according to the Web of Science (WoS), 489 bibliometric reviews have been published on COVID-19, most of them focused on the biological, pharmacological, nursing or medical aspects of the impact of the disease or its treatment. Another considerable part focused on the impact of the pandemic from the point of view of business or the economy. Of these 489 reviews, 17 included some reference to stress [60,61,62,63,64,65,66,67,68,69,70,71,72,73,74,75,76] but none mentioned burnout. Likewise, 5 considered teachers [71,76,77,78], with 3 of them focusing on university teachers [71,77,78,79,80] and the other 2 on the use of learning through ICT [77,80]. None of the available reviews analysed the role of stress and/or burnout in teachers.

Thus, given the importance of burnout and its possible increase due to COVID-19, especially in the case of teachers, together with the discovery of the absence of bibliometric studies on the subject, the present research takes on importance.

The present study aimed to carry out a bibliometric analysis of articles published on the Web of Science (WoS) related to teachers’ burnout and/or stress in a pandemic situation.

Bibliometric studies, unlike other types of systematic reviews, allow a more detailed quantification of scientific production, offering information that may also be present in other types of systematic reviews (such as authors, universities or journal production). However, these other methodologies lack certain useful types of information such as the impact of all these factors, which can be addressed by including the analysis of the number of citations received, the analysis of co-authorship or co-citation networks and a thematic analysis based on the frequency of occurrence of terms and their relationships.

The data provided by this study can provide a global picture of the scientific impact and can facilitate decision-making when establishing policies, promoting innovation plans or allocating resources to mitigate the effects of the pandemic in the teaching environment. Thus, the quality of life and health of teachers, and therefore also the quality of teaching in general, is ultimately improved.

It is this approach that generates the main question of this research. How important is the stress experienced by teachers and how does it affect their health? In addition, the following questions have guided the design of this study:

RQ1—How have the articles published on stress and burnout in teachers evolved since the beginning of the pandemic?

RQ2—Which authors have published the most articles on teacher stress and burnout during the pandemic, and which have been the most cited?

RQ3—Which countries, academic journals and institutions have published the most on the topic of teacher stress and burnout during the pandemic, and how many citations have they received? 

RQ4—Which co-authorship networks, cross-country cooperations, co-citations and co-words are related to the study of teacher stress and burnout in pandemic situations? 

RQ5—What are the main topics studied within this research field?

The article is organized as follows. First, the methodology is presented, with details of the bibliometric methodological techniques and the software used in this study. Next, the results are presented by means of bibliometric tables and maps. The results are then discussed. Finally, the conclusions, the implications for future research and the limitations are presented.

## 2. Materials and Methods

### 2.1. Data Collection

Bibliometric analysis uses bibliographic indicators to analyse the most critical literature in a specific field of research [81]. This study analyses all published articles indexed in the Web of Science Core Collection™ (SSCI, SCI Expanded) on stress and burnout among teachers in pandemic situations. Only Web of Science (WoS) publications were considered, as this is considered the most widely accepted database for the collection and analysis of scientific articles [82].

An advanced search was performed in the subject search field, referring to the title, abstract or keywords of the papers. The search string used in the subject field was: 

((TS = (stress)) OR TS = (burnout OR burn-out OR “burnout syndrome” OR “burn-out syndrome” OR “burn-out syndrome” OR “burn out syndrome”)) AND TS = ((teach* OR school) AND (pandemic OR COVID 19 OR COVID-19 OR Coronavirus OR “Health Crisis” OR “sanitary crisis” OR “healthcare crisis” OR “health emergency” OR “SARS-CoV-2”)).

The search chain was limited by the notification of the first cases of atypical pneumonia by the Chinese authorities to the WHO, which took place on 31 December 2019 [27], up to 29 November 2021, the date of the first case of the Omicron variant. The WHO first reported this variant, B.1.1.529, on 24 November 2021, with the first known case of infection in South Africa reported on 9 November 2021 [83]. Omicron provided a new perspective in dealing with the pandemic, which together with the full vaccination schedule for those aged 12 years and older led to a significant decline in the cumulative incidence, hospitalisation and ICU occupancy [84].

The authors adopted the PRISMA (preferred reporting items for systematic reviews and meta-analyses) approach [85,86] to review and select papers in the literature search. This approach has also been used in other bibliometric studies in various fields [87,88,89].

The search generated 433 articles that were reviewed by reading the title, abstract and keywords to identify related contributions and to select those relevant for inclusion in the analysis.

In this first review, we eliminated duplicate records (N = 36) and checked unknown data and authors’ names in order to avoid misspellings of names and initials. We also addressed synonyms and homonyms in authors’ names through other fields such as the author’s address [90]. As not all co-authors’ addresses are listed in the WoS database, Google was used to complete the information. In the event of changes in the institution, the most up-to-date one was chosen [91]. A further 250 articles were eliminated, based on their suitability according to the following inclusion criteria: (1) literature reviews and empirical studies, (2) scientific journal articles, (3) published in any language in the last 5 years, (4) in the Web of Science core collection SCI Expanded and SSCI databases, and (5) examining stress and burnout coped with by teachers and professors during the health crisis produced by COVID-19. The study was therefore limited to research articles in the strict sense, including only original papers and excluding editorials, book reviews, conference abstracts, letters, editorials and news items. This led us to eliminate 286 articles that did not meet the criteria.

As a result, 147 articles were selected, and the following exclusion criteria were applied: (1) university professors and (2) interventions outside the school setting. A complete reading of the articles led to the exclusion of 72 articles, leaving 75 articles for the following analysis (Figure 1).

### 2.2. Bibliometric Analysis

Duplicate and unrecognised records were identified and homogenised after the plain text data were available. The analysis was subsequently carried out in two different stages. First, the basic bibliometric indices (number of articles published per year, per author, per country, per institution and per journal) were calculated using the HistCite statistical software package (version 2010.12.6; HistCite Software LLC, New York, NY, USA) [92]. Secondly, analysis of co-authorships, cross-country collaborations and co-keywords and the thematic analysis were performed using the R package bibliometrix and VOSviewer.

The data collected for authors, years, countries, journals and cited references were obtained using HistCite (version 10.12). This software presents the information in an orderly and detailed manner. This software performed the basic bibliometric analyses: the number of articles per year, the number of articles per author, the number of articles per journal and the number of articles per country. However, Hitscite shows not only quantitative indicators but also quality indicators: the total global citation score (TGCS) and the TLGCS (local global citation score). For this reason, both quantitative and quality indicators were taken into account in this work. The total global citation score (TGCS) refers to the total number of citations received by the articles selected in the analysis carried out using WoS. The total local citation score (TLCS) represents the number of citations in the WoS received only by the articles selected in the specific analysis carried out.

VOSviewer software [93] was then used to analyse the bibliographic linkage and thematic analysis. The retrieved data were also analysed using R v.3.4.1 software with the R package bibliometrix (http://www.bibliometrix.org, accessed on 15 December 2021) [94,95]. The data were imported into R and converted into a bibliographic data framework. Bibliometrix covers the entire workflow, while VOSviewer facilitates bibliographic coupling analysis and enables identification of significant articles in WoS subject searches, contributing to the bibliometric analysis.

The R software package [95] was used to analyse the basic information of the search string performed, the cross-country collaboration index, the map of cross-country collaborations, the authors’ word cloud and the analysis of strategic diagrams. The strategy diagrams based on the joint word analysis enabled identification of the main research topics, as well as emerging themes in the field.

A literature linkage analysis was also carried out to identify the different clusters. Bibliographic linkage measures the similarity between two articles by identifying the number of references they have in common.

Moreover, the number of references cited in the articles does not change over time. As a result, unlike co-occurrence analysis, this analysis is not influenced by the time at which it is performed. For this reason, it is appropriate for use in systematic literature reviews and has been used in previous studies (Viner et al., 2020 [9]).

## 3. Results

After reviewing all the documents, the WoS database search retrieved a total of 75 articles published in 33 journals by 307 authors. The average number of citations per document was 4.12. A total of 212 keywords and 264 author’s keywords were found. Finally, there were around 4 authors per paper, and the collaboration index was 4.31. This information is presented in Table 1.

### 3.1. Basic Indicators

This first section of the results presents the basic indicators, with the papers and citations per year and the number of papers and citations per author, per institution and per country. For the journals, details are given of those that have published at least 1 article, the number of publications, citations and the impact factor. Finally, the evolution of the authors’ keywords according to the year of publication is presented.

#### 3.1.1. Years

The number of published articles was 75, focusing on the years 2020 and 2021. The search was conducted from 31 December 2019 to 29 November 2021. The first article was published in 2020, and 12 publications were found for that year. There was a significant increase in the number of publications (63) on this topic in just one year. The global citations amounted to 222 in 2020 and 151 in 2021.

#### 3.1.2. Authors

A total of 307 researchers published at least 1 article on the topic of burnout and/or stress in teachers.

The researchers with the most publications on this topic were Lee J and Liu F, Mondragón NI, Ozamiz-Etxebarria N, Pressley, Santamaria MD and Santxo (N = 3). Those with the most global citations were MacIntyre, Gregersen and Mercer (TGCS = 63), followed by Kim LE and Asbury (TGCS = 45) and Mondragon, Ozamiz-Etxebarria, Santamaria and Santxo (TGCS = 25). The results, showing the authors with the most publications are presented in Figure 2, establishing 3 or more documents as the cut-off point. A comparison is also shown for Recs, the TLCS (local citation score) and the TGCS (global citation score).

The following shows the area of training of the authors with the highest number of publications, based on their affiliations. This information is presented in Table 2.

#### 3.1.3. Institutions

As can be seen in Figure 3, and with 2 publications as the cut-off point, Christopher Newport University and Columbia University were the universities with the most published documents, with 3 documents each. However, in terms of global citations and with twelve citations as the cut-off point, the American University of Sharjah and Cape Breton University were the universities with the most global citations (TGCS = 63), followed by the University of York (TGCS = 45), Bucharest University (TGCS = 24), University of the Basque Country (TGCS = 22), Christopher Newport University (TGCS = 13) and the Pontifical Catholic University of Valparaíso (TGCS = 12).

#### 3.1.4. Countries

Researchers from a total of 35 countries published at least 1 article on this research topic. In terms of the number of publications, and with 5 articles as the cut-off point, the country that published the most was the United States (N = 27), followed by China (N = 13), with Spain in third place (N = 12). In fourth place for publications in WoS was Australia (N = 9), followed by the United Kingdom (N = 6) and Italy (N = 5). These figures can be seen in Figure 4.

The map below shows the same distribution by country, showing all the countries that published at least 1 article (Figure 5).

Figure 6 shows that the country that received the most citations in the entire WoS, establishing a cut-off point of more than 40 articles, was the United States (N = 68), followed by Austria and the United Arab Emirates (N = 63). Next were Canada and Spain (N = 60 in both cases), followed by the United Kingdom (N = 48).

#### 3.1.5. Journals

A total of 47 journals have published at least 1 article on this topic. Of all these journals, 8 have published 2 articles and 5 have published more than 2 articles. This number was established as the cut-off point (Table 3).

The leading journals in terms of number of articles published were *Frontiers in Psychology* (N = 14) and *International Journal of Environmental Research and Public Health* (N = 13). Finally, *School Psychology* was ranked third (N = 8), followed by *Frontiers in Psychiatry* and *School Psychology Review* (N = 4).

With regard to the impact factor of the 5 journals that published the most articles, *Frontiers in Psychiatry* had the highest impact factor (JCR = 4.16; Q1), followed by *International Journal of Environmental Research and Public Health* (JCR = 3.39; Q2) and, in third place, *Frontiers in Psychology* (JCR = 2.99; Q2). The results can be seen in Table 3.

#### 3.1.6. Most Common Keywords

There were 276 most common keywords used in the publications studied, with a cut-off point of 3 or more, as presented in Figure 7. The central terms of burnout and stress (N = 25) stand out, followed by impact (N = 17), health (N = 12), mental health (N = 13) and self-efficacy and work (N = 10). Anxiety and depression are also prominent (N = 9), as is job-satisfaction (N = 7).

### 3.2. Co-Citation Analysis

This section presents the co-citation analysis. First, the co-authorship network is represented, followed by the cross-country collaboration networks and finally the keyword networks. All these results are represented in the following maps.

#### 3.2.1. Co-Authorship

Of the total of 257 authors, only collaborations between authors who wrote 1 or more articles together are presented. The 7 co-authorship networks, involving 22 researchers who published a joint article on this topic, are presented. Each of the networks is presented using Louvain’s algorithm. In specific terms, there is 1 network of 5 researchers, 2 networks of 4 researchers, 1 network of 3 researchers and 3 networks of 2 researchers. Figure 8 shows the various collaborative networks.

#### 3.2.2. Collaborations between Countries

As shown in Figure 9 and Figure 10, the United States was the most collaborative country in terms of inter-country collaborations, followed by China and the United Kingdom. There was also a strong network of collaboration between Spain and Chile and between Austria and the United Arab Emirates.

Figure 10 below shows the collaboration networks between countries, with the size of the letters and the width of the connections indicating the number of connections between them.

### 3.3. Thematic Analysis

Finally, this third section presents the results of the thematic analysis. First, we show the bibliographic coupling analyses for both documents and words, and second, we show a strategic diagram of the various themes. All these results are represented in maps.

#### 3.3.1. Bibliographic Coupling for Documents and Keywords

A cut-off point of at least 6 citations per document was established in the bibliographic coupling for documents. Subsequently, only those that were connected were selected, leaving 18 documents in the final analysis, which were distributed over four different clusters (with a different colour for each cluster). The size of the letters is proportional to the number of citations and to the frequency of connections between them. These clusters can be seen in Figure 11.

A thematic review of each of the clusters is presented below, with the number of papers in the cluster, citations and leading authors.

**Red cluster** (210 citations, 9 papers): psycho-emotional state of teachers and coping strategies.

This is the largest cluster and is composed of 9 papers. It received a total of 198 citations. The subject of these papers is mainly related to the psychological state (or mental health or emotional situation) of teachers during the COVID-19 crisis (resulting from work demands and available resources), with some authors measuring stress, anxiety and depression, as well as somatic outcomes (such as insomnia, tension and irritability) and emotional exhaustion (due to stress triggers such as the need to apply new technologies to teaching as a result of school closures).

Some authors suggest the use of techniques such as mindfulness to improve the emotional state of teachers, as well as increasing resources, promoting training or improving professional identity. In other cases, protective factors are identified, as well as the main risk factors. Based on the analyses in these articles, it is important to provide high-quality educational support and services to help maintain teachers’ psychological well-being, including preventive interventions, adequate planning to improve communication and timely legislation.

The most cited article was by Macintyre et al. [97], with 62 citations. These authors examine the stress and coping strategies of a group of teachers during the pandemic, based on a survey measuring stressors and 14 coping strategies grouped into two major constructs: the approach and the avoidant groups. In the former, according to Macintyre et al. [97], the strategies perform active work by acting to change the stressor or accept its presence, and the avoidant strategies present responses such as denial, distraction or repression.

Positive psychological outcomes (well-being, health, happiness, resilience and growth) during the pandemic correlated positively with approach strategies and negatively with avoidant coping, which only correlated consistently with negative outcomes (stress, anxiety, anger, sadness and loneliness).

The second most cited article in this cluster was by Kim and Asbury [4], with 45 citations. This article discusses various emotional aspects resulting from the stress experienced by teachers due to school closures and the need to adapt to the new online context. They address issues such as uncertainty, professional identity, lack of contact with students and their concern for the most vulnerable students as triggers for emotional exhaustion in teachers.

Finally, the article by Matiz, Fabbro, Paschetto, Cantone, Paolone and Crescentini [98] appears in third place with 25 citations. The authors deal with meditation techniques such as mindfulness to improve the mental health or emotional situation of teachers by mitigating the negative effects that they experience in their psychological state due to the public health measures that had to be adopted because of the pandemic. This article measures levels of distress and emotional exhaustion, and assesses mindfulness, empathy and psychological well-being skills.

**Green cluster** (55 citations, 5 papers): technostress and its impact on the working environment.

In second place is the green cluster, which was composed of 5 papers that received 55 citations. This cluster identifies the use of new technologies and the difficulty for teachers in adapting to this new work context as one of the most important stressors. This situation and the need to learn to use ICT led to an increased workload, a mismatch between demands and resources and an incorrect perception of their own skills, triggering a series of negative emotions that led teachers to suffer burnout and technostress.

In this cluster, the most cited article, with 24 citations, was by Panisoara et al. [99]. The authors discuss the role of the use of new technologies as the main occupational stressor in the pandemic era and look at how motivation and attitude can help teachers to cope with working remotely. The article notes a link between motivation and teachers’ intentions to continue with online teaching.

The second most cited article was by Zho and Yao [100]. This received 12 citations and discusses stress as a mental health problem that increased during the pandemic period, leading to anxiety, depressive symptoms, fear, denial and anger. The authors argue that social support can act as a protective factor in reducing stress. If teachers’ psychological distress is not alleviated, their mental health may be affected, and this may in turn impact on the mental health of their students.

Finally, Penado-Abilleira et al. [101], with 7 citations, discuss the adaptation of teachers to the technostress experienced due to the irruption of the online methodology in Spain.

**Blue cluster** (27 citations, 2 papers): the influence of socio-economic and cultural factors on teachers’ quality of life.

In third place, the blue cluster was composed of 2 papers with 27 citations, with the main theme of the influence of socio-economic and cultural factors on teachers’ quality of life during the pandemic and the impact on both physical and mental health.

The most cited article was by Sharma and Bhaskar [102], with 18 citations. The authors discuss how social, economic and cultural factors interfere with the personal and academic status of teachers and students during the pandemic. The risk of unemployment or inequality in the provision of educational technologies and online platforms are stressors and influence overall well-being or the lack thereof. It concludes by highlighting the need to implement support services to combat pandemic problems that most affect vulnerable or under-resourced populations, and thus combat future outbreaks.

The second article by Lizana et al. [103], with 9 citations, focuses on teachers’ quality of life and how it was affected by the pandemic. It consists of a longitudinal study, with teachers assessed in two stages: pre-pandemic and pandemic. The authors explain how before the pandemic, teachers already had a low perception of their quality of life, but that this perception has undergone a significant change in all the indicators of the evaluation questionnaire used (Short Form 36 Health Survey) as a result of the changes that have taken place since 2020. This decline is attributed by the authors to work overload, feelings of uncertainty, loneliness and fear of the situation.

**Yellow cluster** (14 citations, 2 papers): cognitive resources to generate subjective and psychological well-being and reduce stress.

Finally, the yellow cluster was the least cited, with 14 citations in its 2 papers. The papers propose cognitive–behavioural strategies to improve mental health in the time of COVID-19. They propose promoting the adaptive process and resilience through creativity and general well-being.

The article by Anderson et al. [104], with 7 citations, investigates the importance of increasing teachers’ resilience and well-being in order to overcome the situation and the challenges posed by the pandemic. The authors propose using creativity as a way to improve the feeling of self-efficacy in teaching and to reduce anxiety and the effects of traumatic stress caused by the pandemic. The direct relationship between different aspects of creative resources, general well-being and resilience in the face of adversity is discussed in this article.

The article by Zadok-Gurman, Jakobovich, Dvash, Zafrani, Rolnik, Ganz and Lev-Ari [105], which is cited 7 times, argues for the importance of using cognitive–behavioural therapies to reduce teachers’ stress and burnout. These cognitive restructuring processes increase psychological and subjective well-being, mindfulness and resilience, which help to improve teachers’ mental health.

A bibliographic coupling for co-word networks was then performed, and a group of four clusters of different colours is shown in Figure 12. In both cases, the size of the letters is proportional to the frequency of occurrence of the keyword and to the number of connections between them.

The size of the letters is proportional to the frequency of occurrence and the number of connections between the words. Four main groups of keywords were found. The cut-off point was set at 5 or more occurrences of these keywords, and the total number of words was 31. The first group is composed of 7 words, and refers to “burnout”. We also saw related concepts such as “self-efficacy”, “work”, “job-satisfaction” and “distance learning”.

We observed a second network composed of 7 words referring to the central point of “COVID-19”, with “mental health”, “depression”, “resilience” and “pandemic” as major concepts connected to it.

The third network is composed of 10 words, with “impact” and “health” as central terms, related to others such as “job demands”, “resources”, “motivation” and “personality”.

The last cluster, with 7 words, has “stress” as the central term. It refers to concepts such as “mental health”, “anxiety”, “education” and “lockdown”.

#### 3.3.2. Strategic Thematic Analysis

Finally, a strategic diagram of the thematic area analysed is presented (Figure 13). The size of the spheres represents the number of occurrences of these keywords. The upper right quadrant shows the driving themes, the upper left quadrant shows niche/very specialised themes, the lower right quadrant shows core themes and the lower left quadrant shows emerging or disappearing themes. The themes in the upper right quadrant are “association”, “resilience”, “mental-health” and “depression”, all of which are relevant and well-developed for the structuring of this research field. The themes in the upper left quadrant are “job demands”, “emotional intelligence” and “quality scale”, with well-developed internal linkages but few relevant external linkages, and therefore they are only marginally relevant to the field.

The themes in the lower left quadrant are poorly developed and marginal, representing mainly emerging or fading themes. In this case, “anxiety”, “personality” or “health resources” appear to be close to the middle ground and may become emerging themes due to their centrality. Finally, the themes in the lower right quadrant are essential to this field of research but are still developing. As a result, cross-cutting and general basic themes such as “stress” or “impact” and “burnout” or “self-efficacy” also appear in this quadrant. The thematic analysis shows that in order to obtain better results, a research focus related to “stress impact” and “burnout” could be given to “self-efficacy”, as these are essential themes in this field but are yet to be developed.

## 4. Discussion

This article addresses the literature on burnout and job stress that has emerged from the COVID-19 pandemic and its impact on teachers’ health [100,106]. This is due to factors such as uncertainty, work overload, the use of new technologies, perceived self-efficacy and improvisation, among others [101,107,108].

The aim of this analysis was therefore to examine the articles related to teacher stress and burnout that emerged from the COVID-19 pandemic. The analysis carried out enabled us to identify the issues that attracted the most interest among researchers. As previously stated, although bibliometric analyses have certain similarities with other forms of review, they also have particularities that make them especially interesting when dealing with literature reviews. Some examples are the inclusion of the analysis of citations and the analysis of co-authorship or co-citation networks, as well as thematic analysis based on the frequency of occurrence of terms and their relationships.

Although all the existing literature on the subject was published only in the last two years, the interest is evident. The topics addressed, which include stress, burnout, coping strategies, self-efficacy, depression, mental health, impact on work and resources, show a real need in our society and in the field under study, particularly with regard to teaching practice [97].

The results obtained will allow us to evaluate management strategies and identify the most important issues for designing future improvements. We discuss the questions guiding the present research on the publication of articles below, in terms of the topics covered, authors, academic journals, countries and institutions, co-authorship networks, cross-country cooperation networks, co-citations and, finally, the main themes in this field of study.

RQ1—How have the articles published on stress and burnout in teachers evolved since the beginning of the pandemic?

There were 153 articles found on this topic, which were reduced to 75 after the selection and identification of sources by means of a flowchart. These showed that there was interest among researchers on the stress and burnout suffered by teachers prior to the COVID-19 pandemic, but the number of articles on the subject increased significantly after the pandemic and following the measures that were adopted worldwide to help maintain social distancing and reduce infections [4,7,8,9]. Some authors measured the level of stress and burnout before and after the pandemic, endeavouring to assess both the increase and the factors that influenced it [103]. Other authors performed comparative analyses between different countries [109,110]. Ozamiz-Etxebarria et al. [111] carried out a study to measure the symptomatology of teachers when they returned to the classroom after the lockdown period, using a questionnaire measuring symptoms of anxiety, depression and stress.

Given that the situation analysed was during the SARS-CoV-2 pandemic period and used the terms “pandemic”, “COVID-19”, “Coronavirus”, “health crisis”, “sanitary crisis”, “healthcare crisis” and “health emergency”, the first specific article on the stress and burnout suffered by teachers during the COVID period appeared in 2020. The number of articles published increased, with 12 of the 75 documents analysed published in 2020, rising to 63 in 2021. Likewise, there were 222 and 151 global citations (TGCS) in 2020 and 2021, respectively.

RQ2—Which authors published the most articles on stress and burnout in teachers and which were the most cited?

Most of the authors published only one article on stress and burnout during the COVID-19 pandemic focusing on the teaching profile. A total of 7 authors published 3 articles (Lee, Liu, Mondragon, Ozamiz-Etxebarria, Pressley, Santamaria and Santxo) that received no citations in this specific search. However, of these, 2 (Mondragon and Ozamiz-Etxebarria) were cited in the WoS 25 times (TGCS). This suggests that productive researchers are beginning to take an interest in this subject.

A total of 16 authors published 2 articles on the subject analysed. MacIntyre and Mercer had sixty-three global citations (TGCS) but none in the specific search. Finally, Asbury and Kim had 45 global citations (TGCS) in WoS and 11 on publications on this subject.

Most of the articles studied were published in 2021 (63) and 12 of them in 2020, and therefore the number of citations should be rechecked in future research.

RQ3—Which countries, academic journals and institutions have published the most on the topic of stress and burnout in teachers, and what is the impact factor of these journals?

Nineteen institutions published more than 1 article on the research topic. Christopher Newport University and Columbia University were in first place, with 3 articles each. Various universities published 2 articles, but with practically no citations on the topic (TLCS). However, the American University of Sharjah and Cape Breton University both had 63 global citations in the WoS (TGCS).

A total of 35 countries published on the subject, with the USA being the most productive, with 27 publications, followed by China and Spain with 13 and 12, respectively. Spain was also at a high level in terms of the number of citations in the WoS (TGCS), with 60 citations, which was not far behind the USA with 68.

Finally, the journals that published the most articles on stress and burnout in teachers during the pandemic were *Frontiers in Psychology* and *International Journal of Environmental Research and Public Health*, with 14 and 13 articles published, respectively. Both have a high impact factor ((JCR = 2.99; Q2; JCR = 3.39; Q2). However, *Frontiers in Psychiatry*, with 4 published articles, has the highest impact factor (JCR = 4.16; Q1).

RQ4—Which co-authorship networks, cross-country cooperation networks and co-citations studied stress and burnout in teachers in pandemic situations?

Five co-authorship networks were found. The USA was clearly the country with the most collaborations, followed by China. However, there was a strong collaboration network between Spain and Chile.

RQ5—What are the main topics studied within this research field?

The recurring themes that appeared in most of the articles were stress, burnout, impact and mental health.

There were four main clusters on this topic. Of these, the cluster referring to the psycho-emotional state of teachers and coping strategies was the most frequently cited. Several authors investigated stress levels, establishing correlations between psychological states and different types of coping [97]. The second most cited article investigated the stress-triggered emotional aspects of school closures and the teachers’ need to adapt to new work situations, including the online context [4]. Finally, there is research on the levels of distress and emotional exhaustion arising from the pandemic, assessing mindfulness skills, empathy and psychological well-being [98].

The next article deals with technostress and its impact on the work environment. This is a recurring theme in most of the articles discussed. Panisoara et al. [99] and Penado-Abilleira et al. [101] highlight the requirement to use new technologies as a stressor during this pandemic period, together with how motivation and attitude play a decisive role in coping with this stressor. Zho and Yao [100] also highlight the situation of anxiety, depressive symptoms, fear and denial, among other issues that teachers are facing, and how social support can act as a protective factor.

The next cluster addressed the influence of socio-economic and cultural factors on teachers’ quality of life. In this cluster, Sharma and Bhaskar [102] explain how these influence the most vulnerable and under-resourced populations, including both teachers and students. They discuss the need to provide support services to combat these shortcomings and personal situations. Lizana et al. [103] focus on the loss of quality of life among teachers. They carry out research in two stages: pre-pandemic and pandemic, and although both show a perception of a poor quality of life in this professional profile, the authors show that largely as a result of the overload of work, uncertainty, loneliness and fear caused by the pandemic, the difference is significant.

Finally, the last thematic block mentioned cognitive resources to generate subjective and psychological well-being and reduce stress. Various cognitive–behavioural strategies to improve the mental health of teachers are proposed. Anderson et al. [104] propose the use of creativity to increase the feeling of self-efficacy and thus reduce the effects of stress and increase resilience, while Zadok-Gurman, et al. [105] advocate cognitive–behavioural therapies to achieve this goal.

The strategic diagram shows that for better development in this research field, future studies should focus on “association” or “resilience” and “mental-health” or “depression”, which are fundamental topics in this field and are well-developed, considering that they have a “driving” role within the scientific field researched. Factors such as “depression” and “mental-health” in general, should be the subject of study, as the research points to an increased risk during the pandemic. Working on resilience would act as a protective factor.

Furthermore, this thematic analysis shows that in order to obtain better results, a focus of research could be “stress impact” and “burnout” with “self-efficacy”, as these are essential themes in this field that are not sufficiently developed.

Although the density of the themes of “anxiety”, “personality” and “health resources” is currently low and these themes are weak, they are very close to the intermediate point (centrality), and therefore we can consider them as possible emerging themes and therefore suitable for future research.

Finally, we did not observe any importance in the themes of “job demands”, “emotional intelligence” or “quality and scales”, as although they were highly developed, they were isolated in the scientific field studied.

## 5. Conclusions, Limitations and Implications

In conclusion, there was a higher number of collective authorships compared to individual authorships. Only 5 documents were written by a single author. The most common authorship pattern was 4 authors, and the collaboration index was 4.31. These results show a preference for collaborative work in this field. This favours the creation of research groups and communication between specialists. In this sense, we can appreciate that most of the collaborations identified correspond to authors from the same country.

Likewise, there was a high rate of collaboration between countries, especially between the United States, China and the United Kingdom, followed by Spain and Chile, and Austria and the United Arab Emirates. These collaborations can serve to strengthen inter-institutional cooperation ties and create international research networks that contribute to the generation of knowledge on the subject.

Regarding the productivity index, no significant difference was identified between the leading authors, with 3 being the highest number of publications, therefore showing that this is an emerging topic. In terms of countries, the USA is generating more knowledge on the subject of study, with 27 publications, followed by China and Spain with 13 and 12 publications. However, it should be noted that in terms of the highest number of citations in the WoS as a whole, the USA still stands out but is not significantly different from Spain (68 and 60, respectively).

With regard to the thematic analysis, stress, burnout, coping strategies, self-efficacy, depression, mental health, and impact on work and resources, among others, showed the real need that exists in teaching practice to address these issues. Specifically, it can be observed in the strategic diagram presented. The diagram shows how “stress impact” and “burnout”, together with “stress”, “burnout”, “coping strategies”, “self-efficacy”, “depression”, “mental health”, “impact on work and resources” and “health resources” are essential themes yet to be developed. 

The impact of the COVID-19 pandemic on teachers’ emotional well-being and quality of life is an important challenge that needs to be addressed by both the educational community and society at large [3,103]. The study conducted shows that the stress and burnout suffered by teachers during the COVID-19 pandemic led to a growing interest in this topic and in everything related to mental health, coping strategies and measures that can be adopted to alleviate the effects and improve teachers’ emotional and physical condition [111]. This increase is evidenced by the main bibliometric indicators, which show a considerable increase in the number of publications. A total of 372 researchers have published in the last two years, with Mondragón NI, Ozamiz-Etxebarria and Ozamiz-Etxebarria being widely cited. The most productive universities were Christopher Newport University and Columbia University. The American University of Sharjah and Cape Breton University were the universities in first place for global citations, followed by York University, Bucharest University, the University of the Basque Country and Christopher Newport University, among others. If this is related to the countries that have published, out of a total of 35 countries that have published, the USA was the most productive country with 27 publications, followed by China and Spain with 13 and 12, respectively.

It is true that the teaching profession has always brought with it added stress due to excessive workloads, interpersonal communication problems, insufficient training and job insecurity [112], but based on the bibliometric analysis carried out, it can be seen that as a result of the COVID-19 pandemic there has been an increase that could endanger teachers’ health, leading them to suffer from burnout syndrome and causing an increase in cases of sick leave, absenteeism and poor job performance [99,105,113].

Workload can therefore be considered a stress trigger for teachers. Education authorities should prevent this risk [110]. Teachers reported significant workloads, psychosomatic problems and burnout as a consequence of the pandemic and changes in their work context, highlighting the need for training in the use of new technologies [114,115].

The problems detected relating to professional identity and job satisfaction must be improved by training and by informing teachers of the challenges that may arise in the classroom and offering them practical solutions [116].

As a result, there is an evident need to support teachers psychologically [100,111]) and thus safeguard their emotional health [117]. In this respect, there is an important interest in incorporating measures to reduce the stress associated with the use of technology. These measures should aim to avoid the consequences of technostress at the organisational level, such as absenteeism and reduced performance in technology users, especially as a result of non-use or misuse of technology in the workplace [118].

The analysis conducted therefore shows the need for further study in this area. Our present situation shows that this is not a temporary situation that is going to abate, and therefore, society in general and teachers in particular will have to adapt to and live with these circumstances that affect our quality of life and therefore our emotional well-being, interpersonal relationships, material well-being, personal development, physical well-being, self-determination and rights [103].

The results obtained in this research can provide a starting point for other authors to study recurring themes such as the lack of teacher training in new technologies in more depth and the need to provide strategies of all kinds such as cognitive–behavioural strategies, social support and resources of all kinds (methodological, financial and technological, among others) for work performance.

Relevant information on the distribution of articles in journals, the countries where most literature has been published, co-authorship networks, citations and, most importantly, the most recurrent research topics that have aroused the greatest interest among authors were presented and discussed. All this information will be significant for further research. It should be noted that no articles on bibliometric analysis were found in the WoS on the subject under investigation, and this represents the strength of this research and its limitation at the same time, since the search should be extended to other databases such as EBSCO or Scopus as a future line of study.

Excessive workloads, job insecurity, insufficient training and changes in their working environment were the most important causes of stress suffered by teachers during the pandemic. Together with emotional factors such as fear, exhaustion and professional identity problems, these factors can lead to burnout syndrome, which can cause both physical and psychological health problems and lead to absenteeism and low productivity at work.

The solutions provided by the various authors highlight the need to provide teachers with financial and methodological resources and means, as well as the importance of working on cognitive–behavioural strategies. Technological training for teachers and the reduction of the workload sometimes caused by this lack of training and information is one of the resources on which most of the authors agree.

Therefore, studies such as the one carried out in this article will serve to encourage the implementation of public and private policies necessary to improve the health of teachers and, by extension, the quality of teaching provided and, in conclusion, to improve the performance of their work practice.

A possible limitation of the present study can be found in the time frame analysed, as it was limited to the period between the communication by the Chinese authorities to the WHO of the first cases of atypical pneumonia (31 December 2019) and 29 November 2021, the date of the first case of the new Omicron variable linked to mass vaccination of the population. Nonetheless, we believe that this is valuable information and shows the growing interest in the topic under study as evidenced by the progression in publications worldwide. We will also continue to build on the existing literature and expand it over time.

## Figures and Tables

**Figure 1 ijerph-19-07134-f001:**
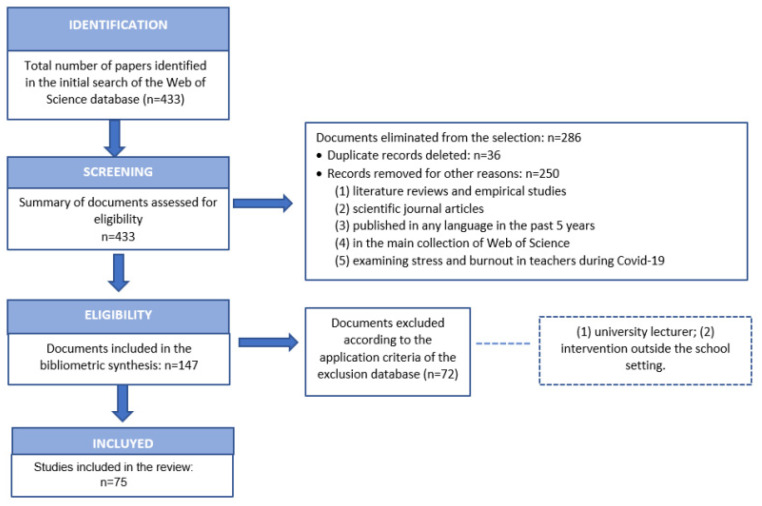
PRISM flowchart detailing the steps in the source identification and selection. **Source:** Page, McKenzie, Bossuyt, Boutron, Hoffmann and Mulrow et al. (2021) [86].

**Figure 2 ijerph-19-07134-f002:**
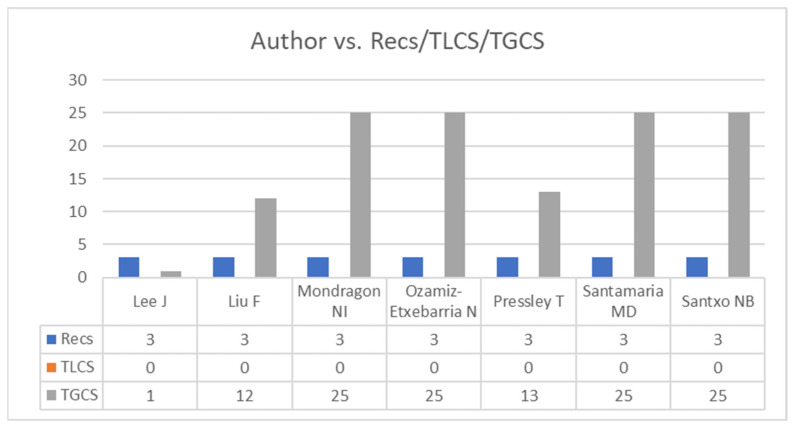
Authors with the most publications (≥3 Recs). Note: Recs is the number of articles, TLCS is the local citation score and TGCS is the global citation score.

**Figure 3 ijerph-19-07134-f003:**
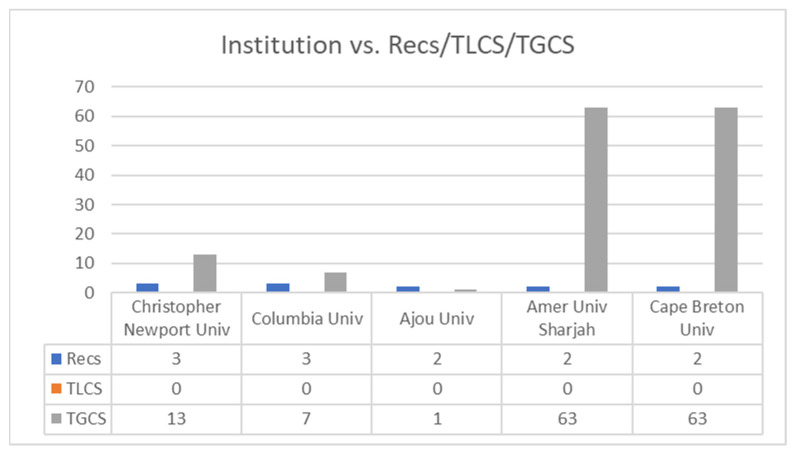
Number of publications by institution (≥2 Recs; ≥12 TGCS). Note: Recs is the number of articles, TLCS is the local citation score and TGCS is the global citation score.

**Figure 4 ijerph-19-07134-f004:**
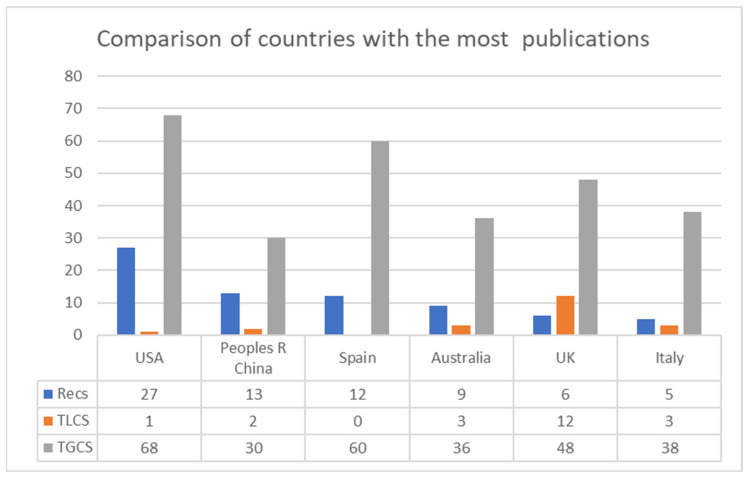
Comparison of countries with the most publications (≥5 Recs). Note: Recs is the number of articles, TLCS is the local citation score and TGCS is the global citation score.

**Figure 5 ijerph-19-07134-f005:**
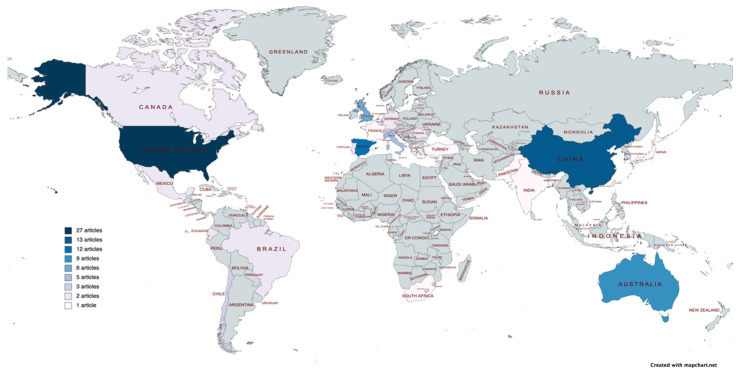
Number of articles published by country (≥1 article).

**Figure 6 ijerph-19-07134-f006:**
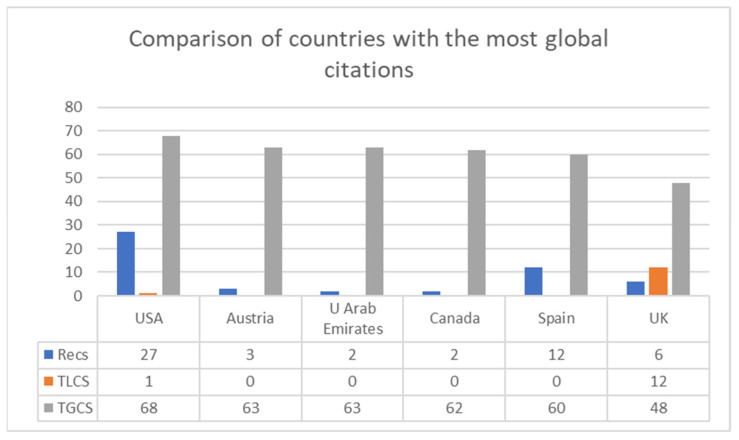
Comparison of countries with the most global citations (≥40 GCTS). Note: Recs is the number of articles, TLCS is the local citation score and TGCS is the global citation score.

**Figure 7 ijerph-19-07134-f007:**
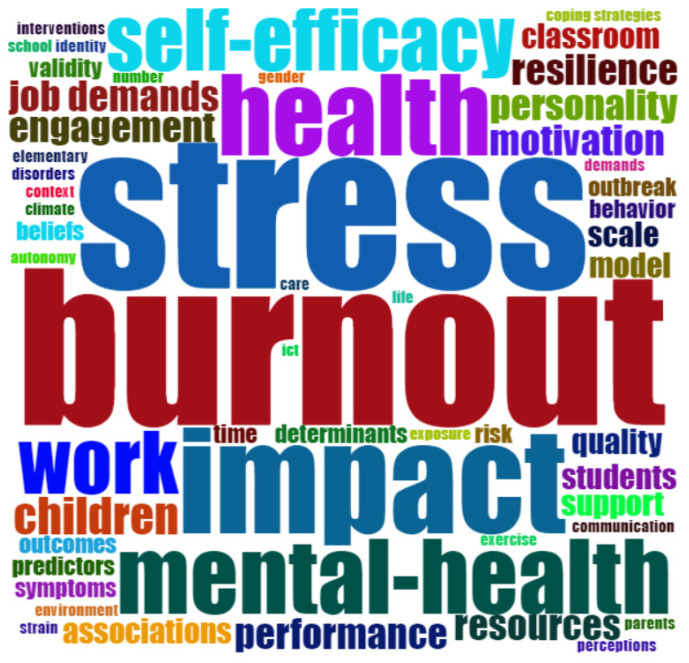
Most frequent words (frequency ≥3).

**Figure 8 ijerph-19-07134-f008:**
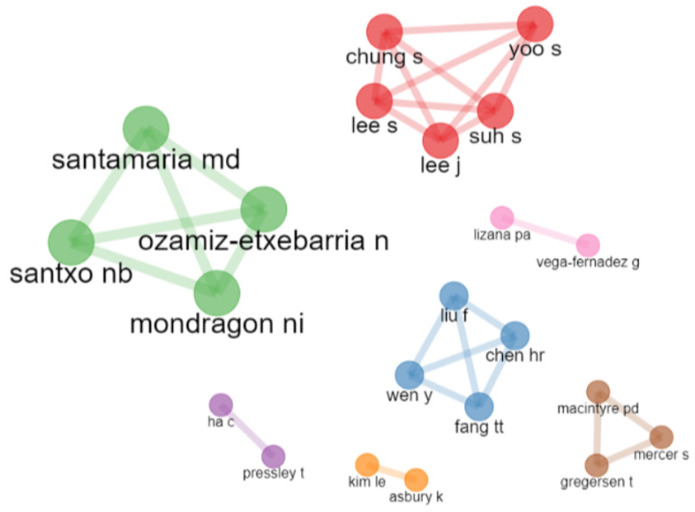
Co-authorship networks (≥1 collaboration).

**Figure 9 ijerph-19-07134-f009:**
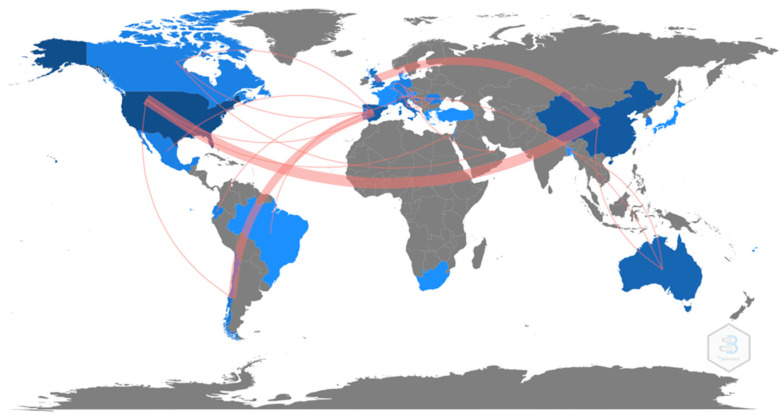
Inter-country collaboration networks map (≥1 collaboration).

**Figure 10 ijerph-19-07134-f010:**
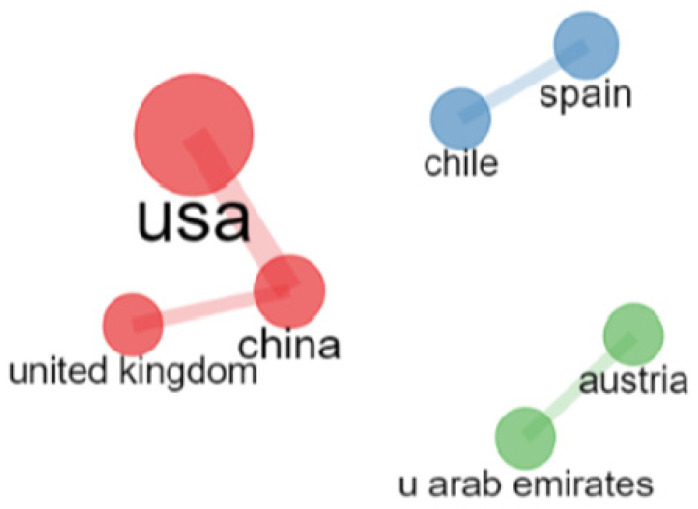
Inter-country collaboration networks (≥1 collaboration).

**Figure 11 ijerph-19-07134-f011:**
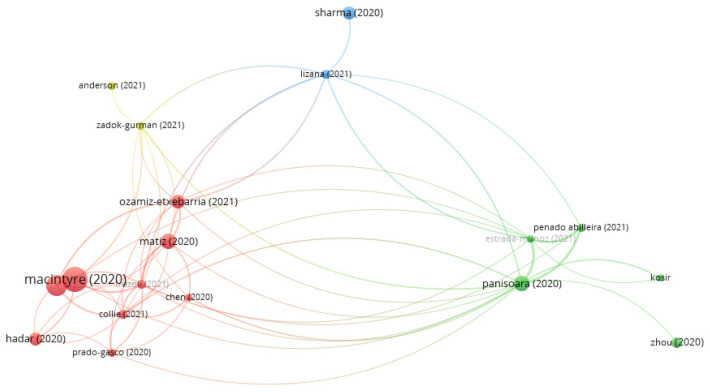
Bibliographic coupling analysis of documents (≥6 citations for publications).

**Figure 12 ijerph-19-07134-f012:**
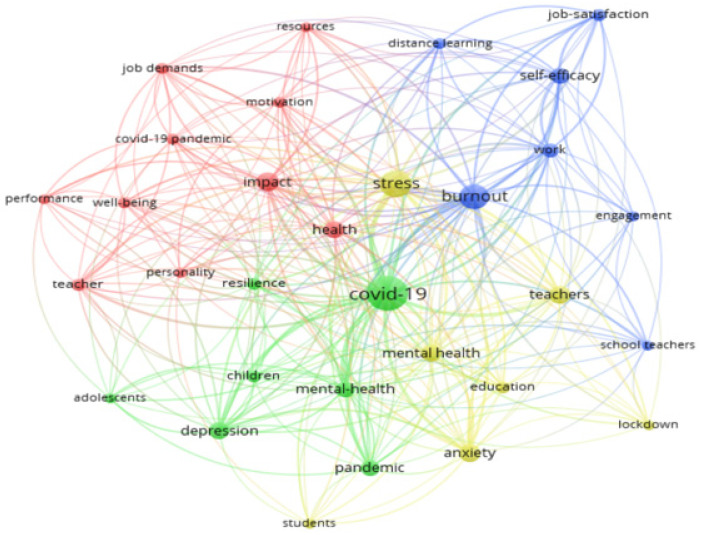
Bibliographic coupling analysis for co-word networks (≥5 co-word networks).

**Figure 13 ijerph-19-07134-f013:**
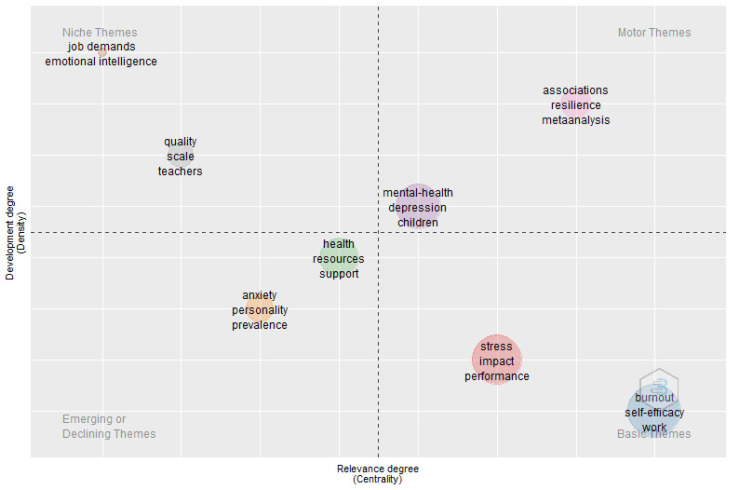
Strategic diagram of stress and burnout in teachers during the pandemic.

**Table 1 ijerph-19-07134-t001:** Summary of information on stress and burnout in teachers during the COVID-19 pandemic.

Main Information about Data	
Journals	33
Articles	75
Average citations per documents	4.97
Avarage citations per year per document	1.99
Avarage years from publication	1.16
References	3947
**Document Types**	
Article	75
**Documents contents**	
Keywords Plus (ID)	212
Author’s Keywords (DE)	264
**Authors**	
Authors	307
Authors Appearances	332
Authors of single-authored documents	5
Authors of multi-authored documents	302
**Authors collaborations**	
Single-authored documents	5
Documents per Author	0.24
Authors per Document	4.09
Co-Authors per Documents	4.43
Collaboration Index	4.31

**Table 2 ijerph-19-07134-t002:** Most productive authors by area of training (≥3 Recs).

Author	Area of Training/Department
Lee J	Psychiatry
Liu F	Educational Science Teacher Education
Mondragon NI	Evolutionary and Educational Psychology Social Psychology
Ozamiz-Etxebarria N	Developmental and Educational Psychology
Pressley T	Psychology
Santamaria MD	Research and Diagnostic Methods in Education
Santxo NB	Didactics and School Organisation

**Table 3 ijerph-19-07134-t003:** Journals by the number of publications and citations received (TLCS and TGCS) and the impact factor (JCR) [96] (≥2 Recs).

Journal	Recs	TLCS	TGCS	JCR (2021)
Frontiers in Psychology	14	0	53	2990
International Journal of Environmental Research and Public Health	13	0	89	3390
School Psychology	8	0	1	4333
Frontiers in Psychiatry	4	0	3	4157
School Psychology Review	4	1	10	2722
Aera Open	2	0	12	2280
Bmc Public Health	2	0	4	3295
British Journal of Educational Psychology	2	11	45	3241
Early Childhood Education Journal	2	1	2	1771
Education and Information Technologies	2	1	3	2917
Frontiers in Public Health	2	0	19	3709
Heliyon	2	0	2	2850
International Archives of Occupational and Environmental Health	2	0	0	3015

Note: Recs is the number of articles, TLCS is the local citation score, TGCS is the global citation score and JCR is the factor impact.

## Data Availability

Data are available upon reasonable request to the corresponding author.

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
