# Peer review of "How Much Do We Care about Teacher Burnout during the Pandemic: A Bibliometric Review"

_ijerph, 2022, doi:10.3390/ijerph19127134_

Round 1
Reviewer 1 Report
General remarks
The aim of the manuscript is to analyze the importance that the literature has expressed about burnout in teachers that occurred during the most recent pandemic crisis. For this, a bibliometric analysis of a selected set of publications in the Web of Science was considered.
Specific remarks
I really enjoyed reading the manuscript, which is particularly interesting from a methodological point of view. Furthermore, I know from experience that teacher burnout was an aspect that should have been given more attention, not only by academia, but also by policy makers in the areas of education and public health.
That being said, let me point out, firstly, some very minor issues to be rectified and, lastly, a, say provoking, suggestion.
In what concerns minor issues:
- Please harmonize the disease designation. I prefer to use "CoViD-19", but obviously both "COVID-19" and "Covid-19" are acceptable, and only one of these should be used in the text;
- Please remove the red underline in "Duplicate records deleted", in Figure 1 (page 4);
- Please correct “The R studio software package [...]”, on page 5, as «bibliometrix» is a package for R and not for R Studio;
- Since section 5. Conclusions not only contains the conclusions, but also, as indeed recommended, the limitations and policy implications of the study, I suggest that it be renamed to 5. Conclusion;
- If so, please remove “Please add” from “Funding: Please add: This research received no external funding” (page 19).
In what concerns the provoking suggestion, it would be very interesting to see what kind of academics, by area of training, have published the most on the subject. In fact, since teacher burnout is a phenomenon that certainly interests psychologists, but also psychiatrists and other types of doctors, but also scientists in the fields of education or public health, or even economists (of education and/or health), perhaps an analysis of this kind -- eventually based on the authors’ affiliations -- would have enriched the study.
Author Response
Dear Reviewer
We have made a careful revision of our article based on the suggestions and comments you have made. We would like to express our sincere thanks for your hard work. Your annotations have allowed us to significantly improve our article and to reflect on future research. We are sending the new manuscript for evaluation.
In order to facilitate your new review, we detail, below, the changes, showing the correction made according to each of your contributions (indicating the page and line whenever possible) as well as the texts added and suggested by you.
We hope that you find the work done correct. If this is not the case, all authors are at your disposal to solve any question or to proceed with new revisions as far as necessary.
Thank you very much for your time and dedication.
Sincerely yours,
The Authors
Report 1
Comment: Please harmonize the disease designation. I prefer to use "CoViD-19", but obviously both "COVID-19" and "Covid-19" are acceptable, and only one of these should be used in the text.
Correction: Following the reviewer's recommendations, the term has been standardised throughout the text. ("CoViD-19")
Comment: Please remove the red underline in "Duplicate records deleted", in Figure 1 (page 4);
Correction: As recommended by the reviewer red underlining has been removed in Figure 1
(p. 4, line 306)
Comment: Please correct “The R studio software package [...]”, on page 5, as «bibliometrix» is a package for R and not for R Studio.
Correction: According to the reviewer's recommendations it was corrected the way of referring to the software R “The R software package […]”
(p. 5)
Comment: Since section 5. Conclusions not only contains the conclusions, but also, as indeed recommended, the limitations and policy implications of the study, I suggest that it be renamed to 5. Conclusion;
Correction: Based on the reviewers' suggestions, the title of section five has been changed to Conclusions, limitations and implications
(p. 19, line 804)
Comment: If so, please remove “Please add” from “Funding: Please add: This research received no external funding” (page 21).
Correction: There has been an error. The research has received funding from one of the participating universities for the publication of the results, so it has been modified in the article.
(p. 21, line 917)
Comment: In what concerns the provoking suggestion, it would be very interesting to see what kind of academics, by area of training, have published the most on the subject. In fact, since teacher burnout is a phenomenon that certainly interests psychologists, but also psychiatrists and other types of doctors, but also scientists in the fields of education or public health, or even economists (of education and/or health), perhaps an analysis of this kind -- eventually based on the authors’ affiliations -- would have enriched the study.
Correction: Following the reviewers' recommendations a table (Table 2) has been included to reflect area of training of the most productive authors.
(p. 7, line 394)
Table 2. Most productive authors by area of training (≥ 3 Recs).

Reviewer 2 Report
First of all, I would like to thank you for the opportunity to read your interesting paper entitled “How much do we care about teacher burnout during the pandemic: A bibliometric review.” This bibliometric review aims to synthesize and explore existing research to identify teachers' burnout and/or stress during the pandemic. The findings show that there are 75 papers from 33 publications, 3,947 cited 15 references and 307 scholars from 35 countries that have published at least one article. The United States ranked first with 16 publications, followed by China and Spain. The United States has the 17 most collaborations, followed by China, and there is also a substantial network of collaboration between Spain and Chile.
However, some concerns in your study need to be addressed. I hope my comments below can be helpful for you as you improve this manuscript to deliver its full potential.
Please justify the need for the study in the introduction. The authors should focus more on addressing what we already know (as we can find, enormous research work has been done on bibliometric studies on Covid 19 and related issues) about the topic before bringing in a gap considering what the paper tries to fill in. This would make it clear to the reader why it is crucial to address the shortcomings in the literature.
After reading the introduction, I wondered why this review is essential and how doing so will address a crucial question in this area. It feels like you got a good solution in your hand--- a bibliometric review, but you appeared not to have a good question. Moving forward, you might want to problematize the literature a bit better and then find a better way to articulate your research contributions.
How is your research, based on a bibliometric review on said topic, different from a meta-analysis or other bibliometric analysis studies?
Recently, I have gone through a few meta-analyses or systematic reviews on the Covid-19. This made me wonder, how is your research different and thus make a unique contribution to the literature? One possible way to address this is that you take a deeper dive into the methods of the review and figure out what this novel method can bring about to deepen our understanding of the development of the literature.
Please verify (https://www.worldometers.info/coronavirus/) the 664,000 deaths in Pakistan and 736,000 in Indonesia.
Could you please explain more about the purpose and benefit of the HistCite statistical software package, R package, Bibliometrics, and VOSviewer? What similarities or dissimilarities among these software-led you to choose all those for analysis.
Why MAXQDA was not considered for thematic analysis?
It would be best if you went a bit deeper into your findings. In many places, you only provided descriptions of results; but you did not give enough explanation of key findings.
The conclusion is a mere synthesis of the research findings. It is underdeveloped, and it falls short in stressing and arguing the original contribution of this research. It should be revised, trying to emphasize the value of this research.
Again, I enjoyed reading your paper and hope my comments can be helpful to you as you improve your manuscript.
Author Response
Dear Reviewer
We have made a careful revision of our article based on the suggestions and comments you have made. We would like to express our sincere thanks for your hard work. Your annotations have allowed us to significantly improve our article and to reflect on future research. We are sending the new manuscript for evaluation.
In order to facilitate your new review, we detail, below, the changes, showing the correction made according to each of your contributions (indicating the page and line whenever possible) as well as the texts added and suggested by you.
We hope that you find the work done correct. If this is not the case, all authors are at your disposal to solve any question or to proceed with new revisions as far as necessary.
Thank you very much for your time and dedication.
Sincerely yours,
The Authors
Report 2
Comment: Please justify the need for the study in the introduction. The authors should focus more on addressing what we already know (as we can find, enormous research work has been done on bibliometric studies on Covid 19 and related issues) about the topic before bringing in a gap considering what the paper tries to fill in. This would make it clear to the reader why it is crucial to address the shortcomings in the literature.
Correction: Following the recommendations of the reviewers, we have tried to better contextualize the interest of the study, especially by reviewing other available bibliometric studies. This text has been added under "Background"
(p. 2, 3, line 124-211)
“Given the importance of burnout and its likely increase due to the situation caused by the emergence of CoViD-19, it is to be expected that there has been a high impact of this topic on the available scientific production in the scientific literature.
In this respect, according to the Web of Science (WoS), 489 bibliometric reviews have been published CoViD-19, most of them focused on biological, pharmacological, nursing, or medical aspects of the impact of the disease or its treatment. Another considerable part has focused on the impact it has had from the point of view of business or the economy. Of these 489, 17 included some reference to stress (citarlos), but none mentioned burnout. Likewise, 5 have considered the group of teachers (cite all but 5), three of them have focused on university teachers, and the other two on the use of learning through ICTs. None of the available reviews have analysed the role of stress and/or burnout in teachers.
Thus, given the importance of burnout and its possible increase, especially in the case of teachers, because of CoViD-19, and the discovery of the absence of bibliometric studies on the subject, the present research takes on importance.
The present study aims to carry out a bibliometric analysis of articles published on the Web of Science (WoS) related to teachers’ burnout and/or stress in a pandemic situation.
Bibliometric studies far from other types of systematic reviews allow a more detailed quantification of scientific production, offering information that may be present in other types of systematic reviews (such as authors, universities, or Journals production. However, these other methodologies lack certain useful information such as the impact of all this by including the number of citations received in the different analyses, the analysis of co-authorship or co-citation networks, as well as thematic analysis based on the frequency of occurrence of terms and their relationships.
The data provided by this study can provide a global picture of the scientific impact and facilitate decision-making when establishing policies, promoting innovation plans, or allocating resources to mitigate the effects of the pandemic in the teaching environment. Thus, ultimately improving the quality of life and health of teachers and therefore also the quality of teaching in general.”
Comment: After reading the introduction, I wondered why this review is essential and how doing so will address a crucial question in this area. It feels like you got a good solution in your hand--- a bibliometric review, but you appeared not to have a good question. Moving forward, you might want to problematize the literature a bit better and then find a better way to articulate your research contributions.
Correction: Following the recommendations of the reviewers, justification has been included in the background (p. 2,3) and the following question has been included
“It is this approach that generates the main question of this research: How important is the stress and burnout experienced by teachers and how does it affect their health? And following questions have guided the design of this study:”
(p. 3, line 209-211)
Comment: How is your research, based on a bibliometric review on said topic, different from a meta-analysis or other bibliometric analysis studies?
Correction: In the preceding comment on the background, the difference between the study and other available studies has been included. After reviewing the literature, there was no bibliometric study that reported on stress and/or burnout in teachers during covid-19.
Comment: Recently, I have gone through a few meta-analyses or systematic reviews on the Covid-19. This made me wonder, how is your research different and thus make a unique contribution to the literature? One possible way to address this is that you take a deeper dive into the methods of the review and figure out what this novel method can bring about to deepen our understanding of the development of the literature.
Correction: A paragraph has been added in the “Introduction” and in the “Discussion” that addresses the issue raised.
Introduction (p. 2,3, line 142-203)
“Bibliometric studies far from other types of systematic reviews allow a more detailed quantification of scientific production, offering information that may be present in other types of systematic reviews (such as authors, universities, or Journals production. However, these other methodologies lack certain useful information such as the impact of all this by including the number of citations received in the different analyses, the analysis of co-authorship or co-citation networks, as well as thematic analysis based on the frequency of occurrence of terms and their relationships.”
Discussion (p.16, 17, line 657-665)
“The analysis carried out enables us to identify the issues that have attracted the biggest interest among researchers. As previously stated, although bibliometric analyses have certain similarities with other forms of review, they have particularities that make them especially interesting when dealing with literature review. Some examples can be the inclusion of the analysis of citations, the analysis of co-authorship or co-citation networks, as well as thematic analysis based on the frequency of occurrence of terms and their relationships”
(p.17, line 671-672)
“The results obtained will allow us to evaluate management strategies and identify the most important issues for designing future improvements.”
Comment: Please verify (https://www.worldometers.info/coronavirus/) the 664,000 deaths in Pakistan and 736,000 in Indonesia.
Correction: We are grateful for the correction. However, this information has been suggested to us by another reviewer for deletion. However, we went to his source and there was indeed a big difference in those two figures. Our source was The Lancet, but there must have been an error. Thank you very much for your input.
Comment: Could you please explain more about the purpose and benefit of the HistCite statistical software package, R package, Bibliometrics, and VOSviewer? What similarities or dissimilarities among these software-led you to choose all those for analysis?
Correction: Although some of the information can be obtained in a similar way with the different programs, each one has certain particularities, for example, VOSviewer is especially valid for the visualization of bibliometric networks. In addition, it performs clusters which are, in turn, very easy to analyze, linking articles, citations, etc. directly, while Bibliometrix allows obtaining multiple types of graphics different from those offered by VOSviewer, such as the evolution map and world map. Histcite is the traditional gold standard for presenting results of basic indicators: articles and citations per year, number of articles and citations per author, per institution and per country, this program is very useful and can extract the results and analyze them according to different parameters.
Since the programs are widely used in the discipline so as not to unnecessarily extend the size of the article, we do not consider it necessary to include this information in the article, although we can include it if the reviewer wishes.
Sources:
Van Eck, N. J., & Waltman, L. (2010). Software survey: VOSviewer, a computer program for bibliometric mapping. Scientometrics, 84(2), 523–538.
Moral-Muñoz, José A.; Herrera-Viedma, Enrique; Santisteban-Espejo, Antonio; Cobo, Manuel J. (2020). “Software tools for conducting bibliometric analysis in science: An up-to-date review”. El profesional de la información, v. 29, n. 1, e290103. https://doi.org/10.3145/epi.2020.ene.03
Wulff Barreiro, E. (2007). El uso del software HistCite para identificar artículos significativos en búsquedas por materias en la Web of Science. Documentación de las Ciencias de la Información, 30, 45 - 64. https://revistas.ucm.es/index.php/DCIN/article/view/DCIN0707110045A
Comment: Why MAXQDA was not considered for thematic analysis?
Correction: Thank you very much for the recommendation. In the study we have included the programs that we have most commonly used and that constitute the standard in the discipline, for future articles we will consider the possibility of using the MAXQDA
Comment: It would be best if you went a bit deeper into your findings. In many places, you only provided descriptions of results; but you did not give enough explanation of key findings.
The conclusion is a mere synthesis of the research findings. It is underdeveloped, and it falls short in stressing and arguing the original contribution of this research. It should be revised, trying to emphasize the value of this research.
Correction: This paragraph has been added in “Conclusions”
(p. 19, line 805-829)
“In conclusion, there is a higher number of collective authorships as opposed to individual authorships. Only 5 documents were written by a single author. The most common authorship pattern is 4 authors, and the collaboration index is 4.31. These results show a preference for collaborative work in this field. This favours the creation of research groups and communication between specialists. In this sense, we can appreciate that most of the collaborations identified correspond to authors from the same country.
Likewise, there is a high rate of collaboration between countries, especially between the United States, China, and the United Kingdom, followed by Spain and Chile, and Austria and the United Arab Emirates. These collaborations can serve to strengthen inter-institutional cooperation ties and create international research networks that contribute to the generation of knowledge on the subject.
Regarding the productivity index, no significant difference is identified between the leading authors, with 3 being the highest number of publications, therefore showing that it is an emerging topic. In terms of countries, the USA is generating more knowledge on the subject of study, with 27 publications, followed by China and Spain with 13 and 12 publications. However, it should be noted that in terms of the highest number of citations in the WoS as a whole, the USA still stands out, but is not significantly different from Spain (68 and 60 respectively).
With regards to the thematic analysis, stress, burnout, coping strategies, self-efficacy, depression, mental health, and impact on work and resources, among others, show the real need that exists in teaching practice to address these issues. Specifically, it is observed in the strategic diagram presented. The diagram shows how "stress impact" and "burnout" with "stress, burnout, coping strategies, self-efficacy, depression, mental health, impact on work and resource" and "health resources" are essential themes yet to be developed, being adequate to obtain better results, to continue under this approach.”

Reviewer 3 Report
A very actual topic, the paper is well written and it includes interesting information. There are a few comments to make:
The abstract needs to be improved, the only information is about the Methods
The first paragraph of Introduction (lines 29-41) is not necessary for the topic of the paper. I suggest to write this part again.
Method is well descripted, congratulations.
Results are amazing in presentation and analysis, really. The idea to present Results and Discussion in clusters is a very good idea, nice job.
Discussion also in very well expressed. The clusters are well connected and the references are actual and well based
In line 604 there are 2 points at the end, I don´t know if it´s a technical mistake or any other thing
There is no part about limitations or biases in the study, maybe it´s a good point to include this part.
Author Response
Dear Reviewer
We have made a careful revision of our article based on the suggestions and comments you have made. We would like to express our sincere thanks for your hard work. Your annotations have allowed us to significantly improve our article and to reflect on future research. We are sending the new manuscript for evaluation.
In order to facilitate your new review, we detail, below, the changes, showing the correction made according to each of your contributions (indicating the page and line whenever possible) as well as the texts added and suggested by you.
We hope that you find the work done correct. If this is not the case, all authors are at your disposal to solve any question or to proceed with new revisions as far as necessary.
Thank you very much for your time and dedication.
Sincerely yours,
The Authors
Report 3
Comment: The abstract needs to be improved, the only information is about the Methods
Correction: Following the recommendations of the reviewers, we have tried to extend the abstract to include the objective of the study, although the limitations of length make it difficult to extend it much further (200 words).
(p. 1, line 14-15, 25-26)
Comment: The first paragraph of Introduction (lines 29-41) is not necessary for the topic of the paper. I suggest to write this part again.
Correction: As suggested by the reviewers the part considered not necessary has been removed, except for a few introductory lines.
(p. 1)
Comment: In line 604 there are 2 points at the end, I don´t know if it´s a technical mistake or any other thing.
Correction: Thank you for your pointing out It was indeed a technical error and has been removed.
(p. 19, line 801)
Comment: There is no part about limitations or biases in the study, maybe it´s a good point to include this part.
Correction: Following the reviewers' recommendations Limitations to this study have been included
(p. 21, line 903-910)
“A possible limitation of the present study can be found in the time frame analysed, as it was limited to the period between the communication by the Chinese authorities to the WHO of the first cases of atypical pneumonia (31 December 2019) and 29 November 2021, the date of the first case of the new Omicron variable together with the complete vaccination schedule in children over 12 years of age. Nonetheless, we believe this is valuable information and shows the growing interest in the topic under study as evidenced by the progression in publications worldwide. We will also continue to build on the existing literature and expand it over time.”

Reviewer 4 Report
The paper describes a descriptive bibliometric analysis of the scientific production of research about burnout/stress in teachers in pandemic situation, targeting the Covid-19 pandemic. The paper is well written and reads well. The content throughout is well presented.
The background gives a good overview of the work related consequences of a stressful work situation, reported by teachers both before and during the pandemic. The background also cover the consequences of burnout and stress. However, the background need more argument describing the research gap and a motivation as to why we need a bibliometric review and also what effects can be obtained by this picture. Arguments for the study should thus, be included in the background and be followed up in the discussion.
Author Response
Dear Reviewer
We have made a careful revision of our article based on the suggestions and comments you have made. We would like to express our sincere thanks for your hard work. Your annotations have allowed us to significantly improve our article and to reflect on future research. We are sending the new manuscript for evaluation.
In order to facilitate your new review, we detail, below, the changes, showing the correction made according to each of your contributions (indicating the page and line whenever possible) as well as the texts added and suggested by you.
We hope that you find the work done correct. If this is not the case, all authors are at your disposal to solve any question or to proceed with new revisions as far as necessary.
Thank you very much for your time and dedication.
Sincerely yours,
The Authors
Report 4
Comment: The background gives a good overview of the work related consequences of a stressful work situation, reported by teachers both before and during the pandemic. The background also cover the consequences of burnout and stress. However, the background need more argument describing the research gap and a motivation as to why we need a bibliometric review and also what effects can be obtained by this picture. Arguments for the study should thus, be included in the background and be followed up in the discussion.
Correction: Following the reviewers' recommendations Arguments have been added in “Background”, “Discussion” and “Conclusion”
Background: (p. 2, 3, line 124-211)
“Given the importance of burnout and its likely increase due to the situation caused by the emergence of CoViD-19, it is to be expected that there has been a high impact of this topic on the available scientific production in the scientific literature.
In this respect, according to the Web of Science (WoS), 489 bibliometric reviews have been published CoViD-19, most of them focused on biological, pharmacological, nursing, or medical aspects of the impact of the disease or its treatment. Another considerable part has focused on the impact it has had from the point of view of business or the economy. Of these 489, 17 included some reference to stress (citarlos), but none mentioned burnout. Likewise, 5 have considered the group of teachers (cite all but 5), three of them have focused on university teachers, and the other two on the use of learning through ICTs. None of the available reviews have analysed the role of stress and/or burnout in teachers.
Thus, given the importance of burnout and its possible increase, especially in the case of teachers, because of CoViD-19, and the discovery of the absence of bibliometric studies on the subject, the present research takes on importance.
The present study aims to carry out a bibliometric analysis of articles published on the Web of Science (WoS) related to teachers’ burnout and/or stress in a pandemic situation.
Bibliometric studies far from other types of systematic reviews allow a more detailed quantification of scientific production, offering information that may be present in other types of systematic reviews (such as authors, universities, or Journals production. However, these other methodologies lack certain useful information such as the impact of all this by including the number of citations received in the different analyses, the analysis of co-authorship or co-citation networks, as well as thematic analysis based on the frequency of occurrence of terms and their relationships.
The data provided by this study can provide a global picture of the scientific impact and facilitate decision-making when establishing policies, promoting innovation plans, or allocating resources to mitigate the effects of the pandemic in the teaching environment. Thus, ultimately improving the quality of life and health of teachers and therefore also the quality of teaching in general.”
Discussion:
(p.16, 17, line 657-665)
“The analysis carried out enables us to identify the issues that have attracted the biggest interest among researchers. As previously stated, although bibliometric analyses have certain similarities with other forms of review, they have particularities that make them especially interesting when dealing with literature review. Some examples can be the inclusion of the analysis of citations, the analysis of co-authorship or co-citation networks, as well as thematic analysis based on the frequency of occurrence of terms and their relationships”
(p.17, line 671-672)
“The results obtained will allow us to evaluate management strategies and identify the most important issues for designing future improvements.”
Conclusion:
(p. 19, line 805-829)
“In conclusion, there is a higher number of collective authorships as opposed to individual authorships. Only 5 documents were written by a single author. The most common authorship pattern is 4 authors, and the collaboration index is 4.31. These results show a preference for collaborative work in this field. This favours the creation of research groups and communication between specialists. In this sense, we can appreciate that most of the collaborations identified correspond to authors from the same country.
Likewise, there is a high rate of collaboration between countries, especially between the United States, China, and the United Kingdom, followed by Spain and Chile, and Austria and the United Arab Emirates. These collaborations can serve to strengthen inter-institutional cooperation ties and create international research networks that contribute to the generation of knowledge on the subject.
Regarding the productivity index, no significant difference is identified between the leading authors, with 3 being the highest number of publications, therefore showing that it is an emerging topic. In terms of countries, the USA is generating more knowledge on the subject of study, with 27 publications, followed by China and Spain with 13 and 12 publications. However, it should be noted that in terms of the highest number of citations in the WoS as a whole, the USA still stands out, but is not significantly different from Spain (68 and 60 respectively).
With regards to the thematic analysis, stress, burnout, coping strategies, self-efficacy, depression, mental health, and impact on work and resources, among others, show the real need that exists in teaching practice to address these issues. Specifically, it is observed in the strategic diagram presented. The diagram shows how "stress impact" and "burnout" with "stress, burnout, coping strategies, self-efficacy, depression, mental health, impact on work and resource" and "health resources" are essential themes yet to be developed, being adequate to obtain better results, to continue under this approach.”
(p. 19, line 903-910)
“A possible limitation of the present study can be found in the time frame analysed, as it was limited to the period between the communication by the Chinese authorities to the WHO of the first cases of atypical pneumonia (31 December 2019) and 29 November 2021, the date of the first case of the new Omicron variable together with the complete vaccination schedule in children over 12 years of age. Nonetheless, we believe this is valuable information and shows the growing interest in the topic under study as evidenced by the progression in publications worldwide. We will also continue to build on the existing literature and expand it over time.”
